# Deciphering Active Prophages from Metagenomes

Kristopher Kieft,[a,b] Karthik Anantharaman[a]

aDepartment of Bacteriology, University of Wisconsin–Madison, Madison, Wisconsin, USA
bMicrobiology Doctoral Training Program, University of Wisconsin–Madison, Madison, Wisconsin, USA

**ABSTRACT** Temperate phages (prophages) are ubiquitous in nature and persist as dormant components of host cells (lysogenic stage) before activating and lysing the host (lytic stage). Actively replicating prophages contribute to central community processes, such as enabling bacterial virulence, manipulating biogeochemical cycling, and driving microbial community diversification. Recent advances in sequencing technology have allowed for the identification and characterization of diverse phages, yet no approaches currently exist for identifying if a prophage has activated. Here, we present PropagAtE (Prophage Activity Estimator), an automated software tool for estimating if a prophage is in the lytic or lysogenic stage of infection. PropagAtE uses statistical analyses of prophage-to-host read coverage ratios to decipher actively replicating prophages, irrespective of whether prophages were induced or spontaneously activated. We demonstrate that PropagAtE is fast, accurate, and sensitive, regardless of sequencing depth. Application of PropagAtE to prophages from 348 complex metagenomes from human gut, murine gut, and soil environments identified distinct spatial and temporal prophage activation signatures, with the highest proportion of active prophages in murine gut samples. In infants treated with antibiotics or infants without treatment, we identified active prophage populations correlated with specific treatment groups. Within time series samples from the human gut, 11 prophage populations, some encoding the sulfur metabolism gene *cysH* or a *rhuM*-like virulence factor, were consistently present over time but not active. Overall, PropagAtE will facilitate accurate representations of viruses in microbiomes by associating prophages with their active roles in shaping microbial communities in nature.

**IMPORTANCE** Viruses that infect bacteria are key components of microbiomes and ecosystems. They can kill and manipulate microorganisms, drive planetary-scale processes and biogeochemical cycling, and influence the structures of entire food networks. Prophages are viruses that can exist in a dormant state within the genome of their host (lysogenic stage) before activating in order to replicate and kill the host (lytic stage). Recent advances have allowed for the identification of diverse viruses in nature, but no approaches exist for characterizing prophages and their stages of infection (prophage activity). We develop and benchmark an automated approach, PropagAtE, to identify the stages of infection of prophages from genomic data. We provide evidence that active prophages vary in identity and abundance across multiple environments and scales. Our approach will enable accurate and unbiased analyses of viruses in microbiomes and ecosystems.

**KEYWORDS** metagenome, microbiome, prophage, software, virus

Address correspondence to Karthik Anantharaman, karthik@bact.wisc.edu.
The authors declare no conflict of interest.

Viruses that infect bacteria and archaea (bacteriophages or phages) are pervasive entities that are ubiquitous on Earth. Phages drive evolutionary adaptation and diversification of microorganisms, play critical roles in global nutrient cycles, and can directly impact human health (1–8). Phages can be organized into two categories according to how they infect a host cell, lytic and temperate. Temperate phages are those that have the ability to integrate their double-stranded DNA (dsDNA) genome

into their bacterial host and can be identified in nearly half of all cultivated bacteria (9). These integrated prophage sequences can coexist with the host cell in a lysogenic stage in which virions are not produced. During host genome replication, the prophage sequence is likewise replicated in a one-to-one ratio. Given host-dependent or environmental cues such as DNA damage or nutrient stressors, or spontaneous activation, the prophage can enter a lytic stage to produce virions and lyse the host (10–15). On the other hand, lytic phages are those that directly enter the lytic stage upon infection with no mechanism for integration and dormancy.

Prophages can affect their host and surrounding microbial communities in both the "dormant" lysogenic stage as well as in the "active" lytic stage. In the dormant stage, prophages can impose physiological changes on the host by altering gene expression patterns, inducing DNA transfer or recombination events, and providing virulence attributes (16–20). For example, the pathogenicity of some strains of *Staphylococcus aureus* is reliant on the presence of integrated prophage sequences (21). In the active stage, the result of phage lysis significantly impacts microbial communities by turning over essential nutrients, especially carbon, nitrogen, and sulfur (22–27). Lysis of bacterial populations likewise alters whole microbiomes by diversifying community structures and expanding niche opportunities (3, 28). For example, the "kill the winner" model of virus population growth suggests that dominant bacterial populations are more susceptible to phage predation, which will facilitate expansions of less abundant taxa as the dominant populations are lysed (29–31). Despite the importance of phage lysis on microbial communities, the proportion of lysis by prophages entering the lytic cycle is unclear. As opposed to strictly lytic phages, it remains difficult to associate prophages with active lysis. This is because prophage genome abundance can fluctuate according to host genome replication in the absence of lysis, whereas lytic phages, with few exceptions, must lyse a host in order to increase the abundance of their genomes.

In addition to traditional approaches such as isolation of phages, advances in high-throughput metagenomic sequencing have sped up the ability to identify a large diversity of lytic and lysogenic phage sequences. Recently developed software has allowed for accurate characterization of prophages in both isolate and metagenomic assembled genomes, namely, VIBRANT (32), VirSorter (33), PHASTER (34), and Prophage Hunter (35). Thus far, this software has allowed us to begin to estimate the total diversity of prophages in nature. However, identifying the genome sequences of prophages does not provide context to their *in situ* state of being in the lysogenic or lytic stage of infection. This information is vital, as it distinguishes which prophage or phage populations are actively impacting a microbial community through lysis events. Moreover, with the exception of Prophage Hunter, current software cannot distinguish prophage genomes that have become "cryptic" or those that have lost functional abilities to enter the lytic stage (36–38). Yet Prophage Hunter still cannot identify if a given prophage is active, only if it may have the ability to become so.

Providing context to the infection stage of a prophage is imperative for accurate conclusions on its role in affecting its host and the microbial community. For example, identifying a prophage encoding a virulence factor or metabolic gene may have important implications for its role in manipulating its host's pathogenic interactions, metabolic transformations, and impacts on nutrient and biogeochemical cycling. In order to place the prophage into context within the microbial community, it would be necessary to first determine which stage the prophage is in, namely, lytic or lysogenic. Assuming that all identified prophages are in a lytic stage could lead to misrepresentations or misinterpretations of the data if the prophage is actually dormant or even cryptic.

Here, we present the software PropagAtE (Prophage Activity Estimator). PropagAtE uses genomic coordinates of integrated prophage sequences and short sequencing reads to estimate if a given prophage was in the lysogenic (dormant) or lytic (active) stage of infection. PropagAtE was designed for use with metagenomic data but can also use other forms of genomic data (e.g., sequence data from isolated microorganisms). When tested on systems with known active prophages, PropagAtE was fully

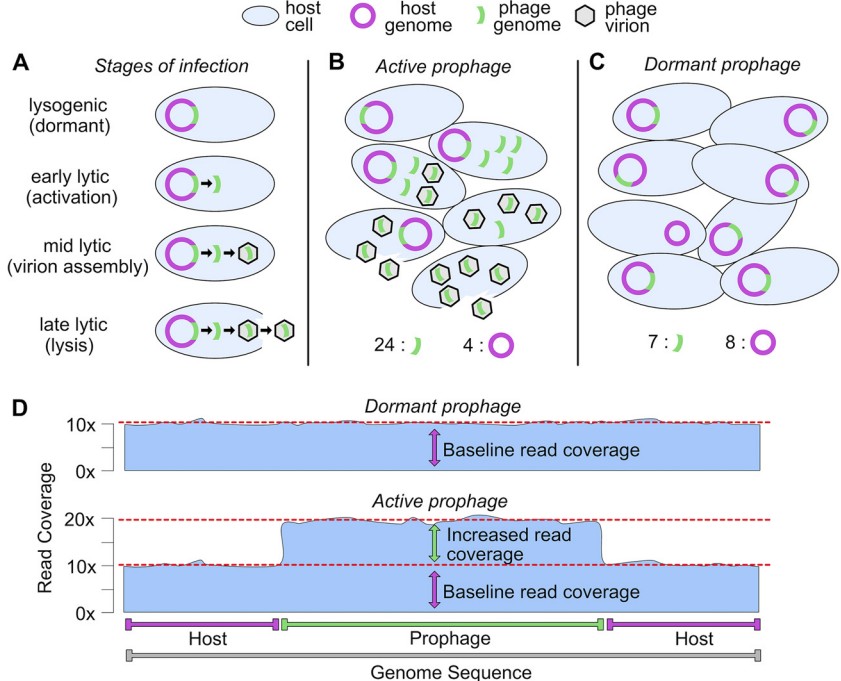

**FIG 1** Schematic conceptualization of PropagAtE mechanism. (A) Stages of integrated prophage infection from the lysogenic (dormant) to lytic (active) stages. Over the course of infection, the prophage/host genome copy ratio increases. (B) Microbial community structure with an active prophage, from phage activation to lysis. The prophage/host genome copy ratio increases to greater than 1:1 through phage genome replication and host genome degradation. (C) Microbial community structure with a dormant prophage in which the prophage/host genome copy ratio is near 1:1. Here, one host is depicted as having cured the prophage from its genome. (D) Conceptual diagram of the read coverage for a prophage in a dormant (top) or active (bottom) stage of infection. Active prophages result in an increased read coverage above the baseline read coverage of the host.

accurate in determining prophages that were active versus dormant, regardless of read coverage depth. No active prophages were identified in control systems encoding prophages that were known to be dormant. PropagAtE was also utilized to identify active prophages in several metagenomes, including the adult and infant human gut, murine gut, and three different peatland soil environments. We show that specific prophages can be identified within differing antibiotic treatment and no-treatment groups of individuals and that activity of those prophages are correlated with particular treatment groups. Finally, we show that identifying the retention of a prophage over time does not necessarily indicate activity over time. PropagAtE is freely available at https://github.com/AnantharamanLab/PropagAtE.

## RESULTS

**Conceptualization of PropagAtE.** Temperate phages that are integrated exist as a component of their host's genome. When the host genome replicates, the prophage is also replicated likewise in a one-to-one ratio. As a result, when sequencing the host genome, the prophage region and the flanking host region(s) are represented equally. Upon activation and entry into the lytic cycle, the prophage sequence is independently replicated for phage propagation and assembly into new virions. At this stage within the host cell, there will be one host genome equivalent for multiple-phage genomes regardless of whether lysis has occurred yet or not. Following lysis, virions containing phage genomes are released into the surrounding environment. These released genomes continue to represent the ratio of prophage to host genome copies if these prophage genomes are still included in the metagenome (Fig. 1A).

The specific ratio of phage to host genomes depends on many factors. One major factor is the burst size of a given phage or the number of virions released from a lysed

host. Phage burst sizes can range from fewer than 10 in the case of crAssphage that infects *Bacteroides intestinalis* (39) to many thousands in the case of phage MS2 that infects *Escherichia coli* (40). Another factor, utilized by many phages, including those that infect marine cyanobacteria, is that the host genome is degraded during the lytic stage to supply nucleotides to the replicating phage genomes, which will further increase the prophage-to-host genome copy ratio (41, 42). Thus, during the lytic stage of phage propagation, as well as postlysis, the ratio of prophage to host genome copies will become skewed in favor of prophage genomes (43, 44). This will lead to a prophage/host genome copy ratio significantly greater than 1:1 (Fig. 1B). If the prophage was in a dormant stage of infection, the prophage/host genome copy ratio would be approximately 1:1 (Fig. 1C). This is likewise dependent on various factors, such as the ability of some members of the host population to "cure" (i.e., remove) the prophage from its genome. Despite nuances in specific prophage/host genome copy ratios, active prophages will yield a ratio greater than 1:1, whereas dormant prophages will yield a ratio near 1:1.

Whether or not the prophage/host genome copy ratio is skewed can be identified using statistical analyses of aligned sequencing read coverage after genome sequencing and read alignment. After sequencing and assembly of a system (e.g., isolated bacterial culture, complex microbiome, etc.), the integrated prophage sequence will assemble as a component of the host genome in an ~1:1 ratio, regardless of activity. However, if a prophage has activated, then the resulting phage genome copies contained in virions are identical to the integrated prophage sequence. Therefore, read alignment to the assembly will recruit reads to the prophage and host regions in a ratio indicative of the stage of infection. During the lysogenic stage where the prophage is dormant, read recruitment will generate even coverage across the regions. Conversely, a prophage that has entered the lytic, active stage will generate an uneven read recruitment skewed toward greater coverage at the prophage region only (Fig. 1D). Read alignment will not determine the true prophage/host abundance, but it can quantify a relative ratio to accurately determine stage of infection.

**Overview of PropagAtE's workflow.** Differentiating active prophages from those that are dormant is essential for accurate representation and evaluation of individual cell- and community-level systems. PropagAtE provides the first automated platform for the identification of active prophages that is scalable for isolate genomes or complex metagenomes. Since most prophages exist as an integrated (i.e., connected) element of a host genome, the read coverage from the prophage and host sections can be compared in a one-to-one manner to estimate a genome copy ratio. PropagAtE utilizes the ratio of prophage/host read coverage along with the ratio's effect size (i.e., significance of the ratio) to designate if a given prophage was dormant or active. The PropagAtE workflow can be simplified into four general steps, data input, read alignment and processing, coverage calculations, and statistical results output (Fig. 2A). Users are given two options for data input, (i) genomes/scaffolds of host sequences with raw short sequencing reads, or (ii) a pregenerated alignment file in SAM or BAM format. If given the former input, reads will be aligned using Bowtie2 (45) to generate a SAM file. All SAM format files are converted to BAM format for more efficient processing (46).

Using the BAM file either generated or supplied by the user, aligned reads exceeding the percent alignment threshold are removed. Following filtering, coverage per nucleotide is extracted, including all nucleotides with zero coverage. To eliminate noise, coverage values at the sequence ends are trimmed off to a length roughly equivalent to the input read length. Then, users are given two options for prophage coordinate data input, direct results from a VIBRANT (v1.2.1 or greater) analysis (32) or a manually generated coordinate file of a specified format. In cases for which multiple prophages are present on a single genome/scaffold, all prophage regions are considered independently. In addition, the host region is segmented to exclude all prophage regions, but each segment is considered a single, cohesive host sequence. That is, if two or more prophages are present on a single host scaffold, neither prophage will interfere

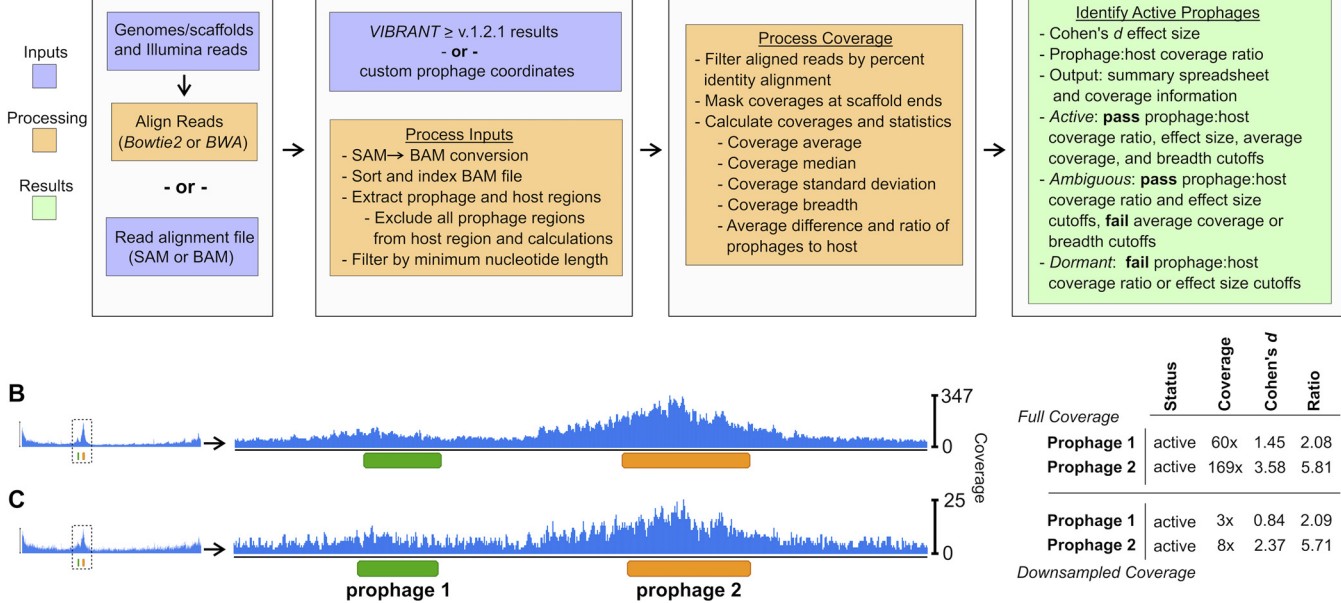

**FIG 2** Workflow and implementation of PropagAtE. (A) Workflow of PropagAtE, including data input, read alignment processing, and results output. Example of read coverage profiles for two active *Bacillus licheniformis* DSM13 prophages with all reads (B) or 5% subsampled reads (C) aligned, respective to the conceptual diagram in Fig. 1D. For panels B and C, statistics for coverage, Cohen's *d* effect size, and prophage/host coverage ratio are shown.

with the other in terms of coverage value calculations, and each prophage is compared to an identical prophage-excluded host region.

For each prophage and host pair, metrics for average coverages, median coverages, coverage standard deviations, and prophage/host coverage ratio are calculated. Each prophage's activity is estimated according to the prophage/host coverage ratio and Cohen's *d* effect size of the coverage difference. Prophages exceeding the default or user-set thresholds for both metrics are considered potentially active. Additionally, potentially active prophages must pass the minimum average coverage and minimum coverage breadth thresholds. If these latter coverage criteria are met, the prophage is estimated to be active; otherwise, the prophage is labeled as ambiguous (Fig. 1A).

**Read alignment can visualize active prophages.** Two activated prophages in the genome of *Bacillus licheniformis* DSM13 (44) were used to visualize active prophage identification using PropagAtE using full and subsampled read sets (Fig. 2B and C). Visualization of the read coverage at each nucleotide in the genome clearly depicted coverage spikes exclusively at the prophage regions. The example prophages existed in close proximity to each other and had differing average coverages (60× and 169×). Both example prophages likewise met the minimum prophage/host coverage ratio (2.08 and 5.81) and Cohen's *d* effect size (1.45 and 3.58) thresholds. These results are in line with the conceptualization of the workflow seen in Fig. 1D apart from notable spikes in coverage at prophage genome centers and host genome ends. The host genome end coverage spikes are commonly explained by the location of the host's origin of replication (47, 48). The coverage spike at the prophage genome center is likely the result of a similar occurrence of a prophage replication-related packaging site (44, 49).

**Positive-control tests for prophages from isolate genomes.** Positive-control tests were utilized in order to set threshold boundaries for PropagAtE to identify active prophages as well as assess the recall rate of PropagAtE. Positive-control samples were considered those for which DNA from both an active prophage and its host were extracted and sequenced in tandem. This method best represents metagenomic samples in which all DNA is extracted and sequenced together. In addition, extraction of both host and free phage DNA together is essential for positive tests because this method will best depict the most accurate prophage/host coverage ratio. Three model systems for which sequencing data were publicly available were identified for use as

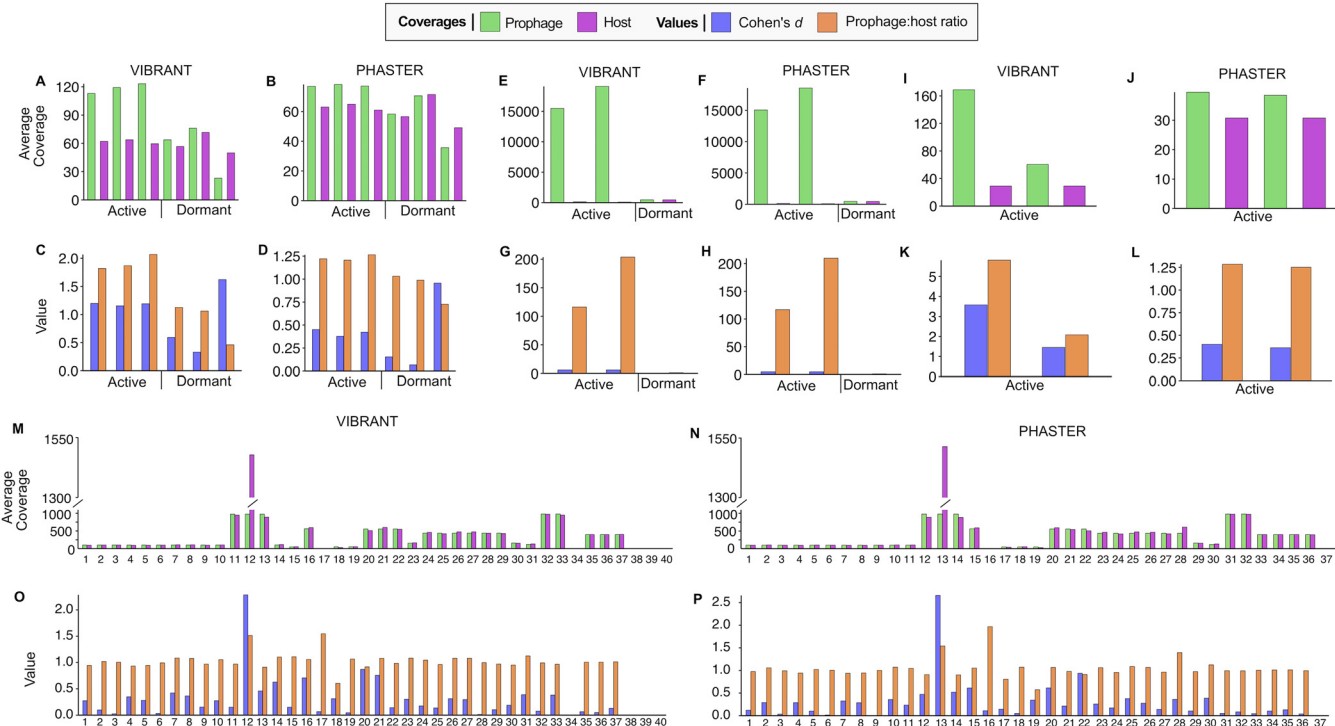

**FIG 3** Positive- and negative-control results using full read sets. (A to L) Positive-control results for *Bartonella krasnovii* OE1-1 (A to D), *Lactococcus lactis* MG1363 (E to H), and *Bacillus licheniformis* DSM13 (I to L). Samples are labeled as containing active or dormant prophages. (M to P) All negative-control results with each value on the *x* axis representing a single prophage. Prophage and host average read coverages (green and purple, respectively), as well as Cohen's *d* effect sizes and prophage/host coverage ratios (blue and orange, respectively), are shown. Each positive- and negative-control set has prophage predictions generated by both VIBRANT and PHASTER (labeled vertically).

positive controls. All experiments and sequencing were performed elsewhere (44, 50, 51) (Table S1 in the supplemental material). Each system, since they represent isolate bacteria, has a much higher read coverage than a typical metagenome-assembled genome. To ensure validation of PropagAtE for both isolate and metagenomic samples, two tests per system were done. One was done with all available reads ("full reads"), and another was done with a random subset of 5% of the reads ("5% reads"). Furthermore, prophages were predicted from these systems using both VIBRANT and PHASTER to ensure accurate predictions despite variable prophage coordinate predictions. All PropagAtE results for positive-control tests can be found in Table S2A.

The first system we tested was *Bartonella krasnovii* OE1-1 and its prophage (50). In triplicate, the bacteria were either induced for prophage using mitomycin C or uninduced as controls. For the induced prophages, the prophage/host coverage ratios were relatively even between the three samples for VIBRANT (1.82, 1.87, and 2.07) and PHASTER (1.22, 1.26, and 1.21). Likewise, in the uninduced control samples. the prophage/host coverage ratios depicted nearly equal coverage (VIBRANT, 1.06 and 1.13; PHASTER, 0.73, 0.98, and 1.03) except one sample from VIBRANT with a low ratio (0.46) (Fig. 3A and B). This suggests the method is reliable across multiple samples or time points for the same phage. The ratio effect size, using Cohen's *d* metric, indicated that the prophage/host coverage ratios observed from the VIBRANT predictions were significant in their difference. For the induced prophages, the effect sizes were greater than 1 (1.20, 1.19, and 1.15), indicating a high dissimilarity between the prophage and host coverages. The uninduced controls' effect sizes were low (0.33 and 0.59) except for the sample with the low ratio, which had a higher effect size (1.62) corresponding to the host having a higher coverage (Fig. 3C). For PHASTER, the same results were not observed. The effect sizes for both the induced prophages (0.45, 0.42, and 0.38) and uninduced controls (0.95, 0.07, and 0.15) were not significant (Fig. 3D). When 5% of the reads were randomly sampled for PropagAtE, the induced and uninduced results were

essentially equivalent to that of the full read set for VIBRANT and PHASTER in terms of prophage/host coverage ratios (Fig. S1A and B) and only marginally lower effect sizes (Fig. S1C and D). This further indicates that high read coverage is not essential, nor significantly impacts, the outcome of analysis. However, this system suggests that the method in which prophages are predicted can determine the outcome and accuracy of PropagAtE activity estimation. Here, VIBRANT predictions yielded expected results, whereas PHASTER predictions yielded dormant predictions where active was expected.

The second system we tested was *Lactococcus lactis* MG1363 and its prophage (51). Similar to the previous system, in one sample, the prophage was induced with mitomycin C, and another was used as an uninduced control. The induction sample was sequenced 1 and 2 h postinduction for a total of two positive samples. For the induced samples, the resulting prophage/host coverage ratios were high and increased over time (VIBRANT, 116 and 204; PHASTER, 117 and 210). In the uninduced control, the prophage/host coverage ratio was, as seen with the previous system, nearly equal (VIBRANT, 1.02; PHASTER, 1.01) (Fig. 3E and F). The effect sizes of the ratio for the induced samples were also high (VIBRANT, 5.98 and 5.91; PHASTER, 4.92 and 4.88), while the effect size of the control sample ratio was low (VIBRANT, 0.10; PHASTER, 0.05). The results from 5% subsampled reads yielded nearly identical equally determinant values for prophage/host coverage ratios (Fig. S1E and F) and effect sizes (Fig. S1G and H).

The third system we tested was *Bacillus licheniformis* DSM13 and its prophages (44). Here, two prophages were spontaneously activated at 26°C, and no control was used for comparison. For VIBRANT, the prophage/host coverage ratios (2.08 and 5.81), as well as the corresponding effect sizes (1.55 and 3.58), were significant (Fig. 3I and K). For PHASTER, the prophage/host coverage ratios (1.74 and 1.28), as well as the corresponding effect sizes (0.93 and 0.40), were not significant (Fig. 3J and L). The same results for both prediction tools were observed when 5% subsampled reads were used (Fig. S1I to L).

Although the available control sample size of the three systems and four unique prophages could not designate a true discovery rate with statistical confidence, the controls tested with VIBRANT predictions yielded high accuracy and recall. Specifically, only the *B. krasnovii* prophage in two induced samples yielded a dormant prediction where active was expected. However, these false-negative results are not entirely unexpected, as the default prophage/host coverage ratio for PropagAtE is set very conservatively to 2.0 and can be reduced to 1.75 while maintaining high accuracy. With a ratio cutoff of 1.75, all controls with VIBRANT predictions would have yielded expected results. When PHASTER predictions were used, the false-negative rate for PropagAtE increased considerably, indicating that accurate prophage coordinate predictions are essential.

**Negative-control tests for prophages from isolate genomes.** Negative-control tests were utilized in order to set threshold boundaries for PropagAtE to identify dormant prophages as well as assess PropagAtE's specificity. Several negative-control samples were used for testing in addition to the control samples presented above. Negative controls were considered those in which a bacterial genome encoding at least one prophage was sequenced in the absence of known prophage induction (i.e., isolate cultures without prophage induction). A total of 19 diverse bacterial genomes encoding 40 predicted prophages by VIBRANT and 37 predicted prophages by PHASTER were used. As before, each system was tested with a set of all reads as well as smaller data set containing 5% randomly subsampled reads. All sequencing was performed elsewhere (Table S1). All PropagAtE results for negative-control tests can be found in Table S2A.

When using the complete reads sets, all prophages were found to be dormant. Average prophage ($1,512\times$ to $0.04\times$) and host ($982\times$ to $0.06\times$) coverages ranged considerably (Fig. 3M and N). All prophage/host coverage ratios were below 1.75 (VIBRANT, max, 1.55; PHASTER, max, 1.54) with the exception of one prophage predicted by PHASTER with a prophage/host coverage ratio of 1.97. However, the effect size of the high prophage/host coverage ratio was only 0.11. All coverage ratio effect sizes ranged from 2.65 to 0.01 (Fig. 3O and P). A total of three prophages predicted by VIBRANT and two prophages predicted by PHASTER had effect sizes greater than 1.75, but the prophage/host coverage

ratios were less than 1.55. For the 5% subsampled read results, the prophage/host coverages ranged from 1.55 to 0, and the coverage ratio effect sizes ranged from 2.14 to 0.01. One prophage from each VIBRANT and PHASTER had an effect size greater than 1.75, but the prophage/host coverage ratios were again less than 1.55 (Fig. S1M to P).

Given that all prophages were identified as dormant, these results suggest that the two metrics, prophage/host coverage ratio and corresponding effect size, function adequately in a check and balance system with each other. Prophages with significantly high prophage/host coverage ratios had insignificant effect sizes and vice versa. Likewise to the positive-control tests, the observed false-discovery rate was zero, though the true accuracy of PropagAtE is likely small but greater than zero. In addition, the negative- and positive-control tests suggest a prophage/host coverage ratio of 1.75, rather than the conservative default of 2.0, can yield accurate results.

**Testing PropagAtE on mock metagenomes.** Sequences assembled from complex metagenome samples typically have lower read coverage than those from isolate systems, and read mapping is performed in the presence of multiple genomes. We next tested PropagAtE on a mock metagenome consisting of prophages predicted by VIBRANT from 21 unique bacteria from the positive- and negative-control tests. *Lactococcus lactis* SD96 from the negative controls was not included in favor of *Lactococcus lactis* MG1363 from the positive controls. A total of 21 corresponding read sets, one per host, were selected, and 300,000, 100,000, or 20,000 paired reads were randomly subsampled per read set and combined to generate the mock metagenome. Thus, three mock metagenomes in total were generated representing 300,000, 100,000, and 20,000 subsampled reads per system (Table S2B). The resulting average read coverages of the prophages was $46\times$, $16\times$, and $3\times$ for the 300,000, 100,000, and 20,000 subsampled mock metagenomes, respectively. The results from the 300,000 subsampled reads mock metagenome corresponded to the results from the positive- and negative-control tests, with 4 active and 36 dormant prophages. A total of 8 prophages with unconfirmed activity status from the positive-control hosts were not considered. For the 100,000 and 20,000 subsampled reads mock metagenomes, the *B. krasnovii* active prophage was identified as dormant due to insufficient prophage/host coverage ratios (1.75 and 1.70, respectively), and all dormant prophages were accurately identified. This depicts that PropagAtE functions well with combined sequences and partial reads from multiple sources, suggesting the method can work suitably with metagenomes.

**Comparing PropagAtE and hafeZ.** The software hafeZ (52) similarly utilizes read coverage to identify active prophages. Contrary to PropagAtE, hafeZ does not take in prophage coordinates as input but, rather, predicts prophages from a host sequence based on read coverage signatures. Using the hafeZ example *Flavonifractor plautii* host genome and prophages predicted by VIBRANT, PropagAtE correctly identified the expected active prophage with a prophage/host coverage ratio of 3.38 and effect size of 5.98. Conversely, hafeZ was unable to identify any prophages in the positive-control data sets presented here. Although PropagAtE and hafeZ cannot be compared directly due to differing methods of identifying active prophages, these results suggest PropagAtE is better capable of identifying more active prophages than hafeZ.

**Applying PropagAtE to identify active prophages in metagenomes.** PropagAtE was designed to rapidly assess the activity of prophages in metagenomes in a high-throughput manner. Additionally, PropagAtE can also identify active prophages in genomes of cultivated organisms, irrespective of the manner of prophage induction (i.e., spontaneously or experimentally induced). To validate the broad utility of PropagAtE, we demonstrate its application on 348 metagenomic samples from a variety of environments, adult and infant human gut, murine gut, and peatland soil (22, 53–57) (Table 1 and Table S1). A total of 349 semiredundant prophages were identified as active across all samples. Per sample, the percentage of prophages that were active ranged from 0% to 18% with a combined average of 1.1% (Fig. 4). The murine gut had the most active prophages per sample with an average of 8.9%, whereas all human gut samples had a combined average of 1.1%. With a prophage/host coverage ratio of 1.75, the number of active prophages increased to 402,

**TABLE 1** Summary of metagenomic sample data sets[a]

| Data set | Description | No. of samples | No. of prophages | No. of hosts | Reference(s) | Label |
|---|---|---|---|---|---|---|
| Human gut (fecal) | Adult individuals with colorectal adenoma, carcinoma, or healthy controls ("CRC") | 15 | 489 | 484 | 57 | a |
| Human gut (fecal) | Adult individuals with Crohn's Disease or healthy controls ("HeQ") | 96 | 2,938 | 2,897 | 54 | b |
| Human gut (fecal) | Infant individuals given antibiotics or untreated controls ("infant gut") | 139 | 356 | 333 | 55 | c |
| Peatland (soil) | Peatland soil cores of bog, fen, and palsa environments ("soil") | 75 | 379 | 375 | 22, 56 | d |
| Murine gut (fecal) | Virome fraction samples from the murine gut ("murine gut") | 23 | 1,308 | 1,292 | 53 | e |
| Human gut (fecal) | Time series of adult individuals with Crohn's disease ("IjazUZ") | 12 | 155 | 153 | 58 | f |

[a]The environment type, description of the data set and total number of samples per metagenomic data set are provided. The final column, "Label", corresponds to labeling in Fig. 4 and Fig. S4.

with a combined average of 1.3%. These results show that for metagenomic samples, most prophages identified as integrated into a host genome are dormant or activity is undetectable. All PropagAtE results for metagenomic samples can be found in Table S3A and B.

For metagenome data sets with various conditions (e.g., antibiotic dosage), no significant difference was observed in the total number of active prophages per condition (Fig. S2 and Table S4A). However, utilizing PropagAtE to identify which sets of prophages are active yielded interesting results. For example, hosts with active prophage populations were compared from the gut of infants given antibiotics compared to infants without antibiotics. A total of 62 host populations with a combined 192 active prophages were compared. Interestingly, a distinct pattern was observed wherein prophage activity was correlated with antibiotic treatment per host population. Generally, a host population had prophage activity in either antibiotic treatment or no treatment, with few host populations having prophage activity uncorrelated with a treatment (Fig. 5; Table S4B). This indicates that although a given prophage or host population may be found across multiple samples, they may be predominately active in specific treatments.

**Estimating prophage activity over time.** To further explore the activity of specific prophage populations over time, a sixth set of metagenomic samples was used (58). This set included human gut fecal samples from three different children with Crohn's disease. For each individual, four time series samples were taken at approximately days 0, 16, 32, and 54. Among all 3 individuals, a total of 11 unique prophages were identified across all 4 time points. None of the 11 prophages were shared between 2 or more individuals. Therefore, these 11 prophages were found to be consistently present and retained stably over time. All prophage populations encoded hallmark phage proteins, nucleotide replication proteins, and lysis proteins, indicating they likely have the ability to activate (i.e., not cryptic). For most populations, genes for integration were also identified (Fig. S3A; Table S5). Furthermore, one prophage population encoded the auxiliary metabolic gene *cysH* for assimilatory sulfate reduction, a metabolic process that can yield hydrogen sulfide, which has been implicated in exacerbating inflammatory bowel diseases such as Crohn's disease (59, 60). Another prophage population encoded a RhuM family virulence protein. Yet PropagAtE identified none of these prophages to be active at any time point. This conclusion is important, as it suggests that the prophages, in addition to the *cysH*- and *rhuM*-like genes, were present but may not have been actively impacting the microbial community at the time of sample collection. Genome alignment of each prophage population yielded 99.8 to 100% identity with a maximum number of two nucleotide differences between members of a population (Fig. S3B). The lack of sequence diversification likewise suggests the prophage populations were primarily dormant over time since active phage genome replication typically results in nucleotide changes. However, the minor nucleotide differences may have resulted from alignment or sequencing error or from prophage activity between the time points sampled.

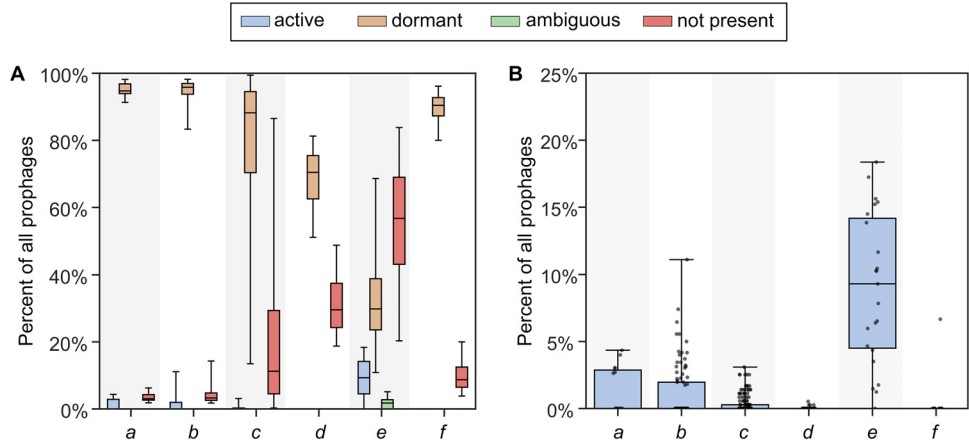

**FIG 4** Percent of prophages by activity category in metagenomic samples. (A and B) Five sets of metagenomic samples are compared with all activity categories (A) and only the active prophage category (B). For panel B, each dot represents a single sample. Identifier labels a to f on the *x*-axis correspond to the final column, "Label," in Table 1.

**Sequencing depth does not correlate with total active prophages.** As a final validation test, we examined if the total number of sequencing reads, as an estimation of sequencing depth, had an impact on the total number of active prophages identified. It may be assumed that since PropagAtE relies on read coverage, samples with a greater number of reads would identify disproportionately more active prophages. Using five of the metagenomic sample sets (Table 1), we correlated the total number of reads used by PropagAtE to the total number of active prophages identified. Four of the five sets of metagenomic samples yielded near-linear, flat trends, indicating no correlation between total reads and total active prophages. The fifth set, representing infant gut samples, depicted more of a trend toward a correlation between more reads and more active prophages. However, the trend was not significant (Fig. S4 and Table S3A).

**PropagAtE run time.** Efficiency and quick run speed are essential for large-scale metagenomic workflows. PropagAtE was designed to meet the needs of these analyses, such as those with many samples or large file sizes. PropagAtE is likewise scalable for smaller data sets. To show this, we estimated the total run time for various isolate and metagenome samples. For isolate samples, run time for PropagAtE analysis was 10 to 90 s with an alignment format file (i.e., BAM format) as the input. For metagenomes, the run time was similar (5 to 45 s) (Table S6). The main factor affecting run time is read alignment performed by Bowtie2, which had run times of 1 to 12 min, depending on input reads and reference genome sizes. It is important to note that the run time for large-read data set inputs significantly improves when utilizing the multithreading feature.

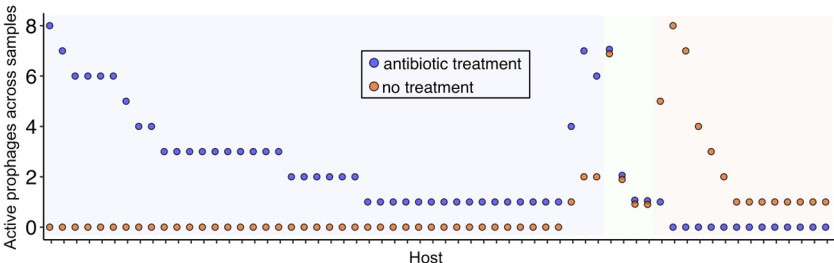

**FIG 5** Active prophages identified in infant gut samples. Each host (*x*-axis) is labeled with two points, one for the total number of prophages identified in antibiotic treatment samples (blue) and one for the total number of prophages identified in no-treatment samples (orange). Background highlighting depicts hosts with proportionally more active prophages in antibiotic treatment samples (blue), more active prophages in no-treatment samples (orange), or equivalent active prophages in both treatment groups (green).

## DISCUSSION

Phages are key contributors to microbiome dynamics in essentially all environments on Earth (6, 25–27, 61–64). With the availability of high-throughput sequencing and newly developed software tools, we have the ability to identify and study these diverse phages (32–35). This includes both strictly lytic phages as well as integrated prophages. However, little emphasis has been placed on identifying which populations of identified prophages are actively replicating as opposed to existing in a dormant or cryptic stage of infection.

Here, we have presented the software tool PropagAtE for the estimation of activity of integrated prophages using statistical analyses of read coverage. Although the concept of using read coverage to predict prophage activity is not new (44), PropagAtE is the first benchmarked implementation of the method into an automated software for use with large data sets, such as metagenomes. PropagAtE functions by quantifying the relative genome copy ratio between a prophage region compared to a corresponding host region. Only prophages that have activated and begun propagation (e.g., genome replication and virion assembly) will yield prophage/host ratios sufficiently greater than 1:1. The prophage/host genome copy ratio, estimated by using read coverage ratios, as well as the ratio's effect size, are used to classify a prophage as active or dormant. We provide evidence to show that PropagAtE is fast, sensitive, and accurate in predicting prophages as active versus dormant and have applied the method to various metagenome samples.

Identifying which prophage sequences are active versus dormant in a sample provides several benefits. Namely, assuming that all identified prophages are active is an overestimation and will lead to a misrepresentation of the *in situ* dynamics of a microbial community. For example, we show here that 11 unique prophages identified in human gut samples from the same individual over time may not necessarily be active when identified. The most accurate representation of the prophages is to conclude that their effect on the resident microbial communities likely occurred at a time point not sampled or that the prophages were consistently dormant. Another benefit includes making accurate conclusions on the role of host bacteria in a given sample. Foremost, prophages can be responsible for the virulence of multiple human pathogens, such as *Clostridioides difficile*, *Clostridium botulinum*, *Staphylococcus aureus*, and *Corynebacterium diphtheria* (65–70). Although some virulence effects are present during prophage dormancy and expression of specific genes, many require activation of the prophage. In addition to virulence, bacteria actively infected by a phage can have a modified metabolic landscape compared to bacteria uninfected or harboring a dormant prophage. Several examples include the phage-directed regulation of sulfur, carbon, nitrogen, and phosphorus metabolism in various cyanobacteria and enterobacteria (23, 71–73). This distinction is vital when assessing the role of the microbial community in an environment. Related to this, activity can provide context to any auxiliary metabolic genes identified on the prophage genome, such as *cysH* for assimilatory sulfate reduction described here. In the human gut specifically, identifying phage-encoded genes for sulfur metabolism may have important implications for the health of the gastrointestinal tract and a phage's role in the manifestation or perturbation of diseases (63, 74). If a prophage encoding an auxiliary metabolic gene is identified, determining the stage of infection of the prophage can provide context to the effect of the auxiliary metabolic gene.

It is important to point out several unavoidable caveats to the implementation of PropagAtE. First, accurate prophage/host genome copy ratio estimations are inhibited if the sample is size fractionated before sequencing. For example, many aquatic samples are size fractionated by filtering onto a 0.2-$\mu$m filter. In these cases, only prelysis infections will be picked up by read coverage because the genomic content present in released virions will likely pass through the 0.2-$\mu$m filter. Second, not all prophages exist as integrated sequences, such as those that are episomal. Prophages that are episomal do not have attached host sequence and therefore cannot have prophage/host read coverage compared in a one-to-one manner and, for metagenomes, cannot have accurate host prediction. This also applies to prophages that do not assemble as integrated components of a host scaffold. However, it is worth noting that for integrated

prophages, PropagAtE functions whether the host region flanks the prophage on one or both sides. Third, though not verified, is that inactive prophages may be more likely to assemble with a host scaffold. Since active prophages lyse their host and potentially degrade their host's genome, more activity of a prophage may lead to a lower probability of assembling as an integrated prophage. Fourth, induction of prophages within a host population may occur asynchronously and lead to consistent activity with low prophage/host coverage ratios, causing activity to be missed. Fifth, some host populations may include some members that encode a prophage and some members that do not. In the latter example, the prophage/host ratio is initially skewed to less than 1, making it more likely for PropagAtE to miss activity. Due to the caveats presented, PropagAtE is intended to be used for identifying active prophage sequences rather than assessing the total number or fraction of prophages that are active in a sample. In this context, PropagAtE performs with little to no observed error. Finally, PropagAtE has been developed and tested using short-read sequencing data and is not yet suitable for long-read analyses.

Overall, our results demonstrate that PropagAtE will facilitate the accurate characterization and study of viruses in microbiomes and nature. Examples of future applications of PropagAtE include the exploration of prophages in human health and disease, detection of environmental and chemical triggers for induction of prophages, phage therapy research (for disqualifying prophages), and in environmental systems research.

## MATERIALS AND METHODS

**Data sets used for control tests.** All data sets, genomes, and reads used for positive- and negative-control tests were acquired from publicly available data sets on NCBI databases (75, 76). See Table S1 in the supplemental material for details of studies and accession numbers. VIBRANT (v1.2.1) (32) and PHASTER (accessed December 2021) were used for identification and annotation of all prophages. Only VIBRANT was used for identification of prophages from metagenomes. For the mock metagenome, reads were randomly subsampled using seqtk (v1.3-r106, sample) (https://github.com/lh3/seqtk).

**Dependencies and equations.** Bowtie2 (v2.3.4.1) (45) was used for read alignment. SAMtools (v1.11) (46) and PySam (https://github.com/pysam-developers/pysam) were used for manipulation, conversion, and reading of SAM and BAM alignment files. To calculate coverage, aligned reads are filtered according to the percent identity alignment, as calculated by subtracting the number of gaps, $g$, and the number of mismatches, $m$, in the alignment from the length of the alignment, $l$, and then dividing by $l$.

$$\text{percent identity alignment} = \frac{l - g - m}{l} \times 100\%$$

Cohen's $d$ metric is used to calculate the effect size of prophage/host coverage ratios. Cohen's $d$ (77) is calculated using the following equation, where $\overline{X}_{\text{host}}$ and $\overline{X}_{\text{prophage}}$ are the average read coverages of the host and prophage regions, and $S_{\text{host}}$ and $S_{\text{prophage}}$ are the standard deviations of the coverages:

$$d = \frac{\overline{X}_{\text{host}} - \overline{X}_{\text{prophage}}}{\sqrt{\frac{S_{\text{host}}^2 + S_{\text{prophage}}^2}{2}}}$$

**Metagenome assembly and analyses.** Metagenomes for the murine gut microbial fraction samples were assembled in this study. Details of raw read sets from murine gut samples used for assembly can be found in Table S1. SPAdes (v3.12.0) (78) was used for genome assembly (–meta -k 21,33,55), and the resulting best scaffold assemblies were retained. The human infant gut and peatland soil metagenomes were assembled previously in their respective studies (22, 55, 56). Both human adult gut metagenomes were assembled by Pasolli et al. (79).

For the human gut time series samples, integrated prophages were predicted using VIBRANT (v1.2.1). To check for integrated prophage sequences that were not assembled with a host scaffold, integrated prophages were compared to all identified phages using dRep (v2.6.2, dereplicate –ignoreGenomeQuality -sa 90 -pa 90) (80). Identical, nonintegrated phage sequences were considered a part of the same prophage population. Genome alignments were performed using progressive Mauve (v1.11, default settings) (81).

**Visualization.** Geneious Prime 2020.1.2 was used for visualization of example read coverage values. R package ggplot2 and Python packages Matplotlib and Seaborn were used for visualization of graphs (82, 83).

**Setting default thresholds for PropagAtE.** PropagAtE has several variable settings and thresholds that can be set by the user, percent identity of aligned reads, masking of coverage values at genome/scaffold ends, minimum prophage/host coverage ratio, minimum Cohen's $d$ effect size, minimum average coverage of the prophage, and minimum breadth of coverage of the prophage. In addition, PropagAtE requires that all prophage and host sequences must each be at least 1 kb in length.

Percent identity read alignment is used for more accurate read alignment processing. This setting is meant to be sensitive for accurate read alignment while allowing for minor errors (default, 97%). Another

coverage metric is masking of coverage values at genome/scaffold ends. This setting is particularly important for metagenomic scaffolds that likely represent partial sequences. For this metric, a generalized length of 150 bp is used to mask (i.e., not consider for calculation) the respective number of coverage values from each scaffold end in order to account for lower coverage values at partial scaffold ends.

The final four settings are used for determination of prophage activity and significance: The most important threshold is the prophage/host coverage ratio, which is set to 2.0 by default and can be reduced to 1.75 for increased sensitivity. The default was selected to be as close to the minimum requirement for designating true active prophages as active in control tests while maintaining a significant gap from true dormant prophages in order to reduce false-positive identifications. Finally, Cohen's $d$ effect size setting is set to 0.70, which falls in the general range of "medium" significance (77). This threshold is useful for contextualizing prophage/host coverage ratios, especially for high-coverage genomes/scaffolds. Again, the default was selected according to control tests for reducing false-positive identifications. The thresholds for minimum coverage (default, 1.0) and minimum breadth (default, 0.50) of prophage regions are used to ensure that only prophages that are likely to be present in the sample (i.e., sufficient coverage) are considered in analyses.

**Data access.** The PropagAtE software and associated files are freely available as a Python package at https://github.com/AnantharamanLab/PropagAtE. All isolate and metagenome genomic sequences and reads used in this study are publicly available; see Table S1 for details. Additional details of relevant data are available on request.

## SUPPLEMENTAL MATERIAL

Supplemental material is available online only.
**FIG S1**, PDF file, 0.1 MB.
**FIG S2**, PDF file, 0.1 MB.
**FIG S3**, PDF file, 0.7 MB.
**FIG S4**, PDF file, 0.1 MB.
**TABLE S1**, XLSX file, 0.02 MB.
**TABLE S2**, XLSX file, 0.1 MB.
**TABLE S3**, XLSX file, 8.7 MB.
**TABLE S4**, XLSX file, 0.01 MB.
**TABLE S5**, XLSX file, 0.03 MB.
**TABLE S6**, XLSX file, 0.01 MB.

## ACKNOWLEDGMENTS

We thank the University of Wisconsin, Office of the Vice Chancellor for Research and Graduate Education; University of Wisconsin, Department of Bacteriology; and University of Wisconsin, College of Agriculture and Life Sciences, for their support. We also thank Z. Zhou, A. Adams, and R. Salamzade for their helpful feedback and discussions.

K.K. was supported by a Wisconsin Distinguished Graduate Fellowship Award and a William H. Peterson Fellowship Award from the University of Wisconsin-Madison. This research was supported by National Institute of General Medical Sciences of the National Institutes of Health under award number R35GM143024.

We declare no competing interests.

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
