## [Reviewer comments · mSystems]

Deciphering active prophages from metagenomes

Kristopher Kieft and Karthik Anantharaman

Corresponding Author(s): Karthik Anantharaman, University of Wisconsin-Madison

Review Timeline:

Submission Date:

January 28, 2022

Accepted:

March 7, 2022

Editor: Robert Beiko

Reviewer(s): The reviewers have opted to remain anonymous.

Transaction Report:

DOI: <https://doi.org/10.1128/msystems.00084-22>

March 7, 2022

Dr. Karthik Anantharaman
University of Wisconsin-Madison
Bacteriology
1550 Linden Dr
4550 Microbial Sciences Building
Madison, Wisconsin 53705

Re: mSystems00084-22 (Deciphering active prophages from metagenomes)

Dear Dr. Karthik Anantharaman:

Your manuscript has been accepted, and I am forwarding it to the ASM Journals Department for publication. For your reference, ASM Journals' address is given below. Before it can be scheduled for publication, your manuscript will be checked by the mSystems production staff to make sure that all elements meet the technical requirements for publication. They will contact you if anything needs to be revised before copyediting and production can begin. Otherwise, you will be notified when your proofs are ready to be viewed.

Corrections:

Lines 272-273: "One prophage from each VIBRANT and PHASTER had an
273 effect sizes greater than 1.75" -> "One prophage from each of VIBRANT and PHASTER had effect sizes greater than 1.75"
L357: "Nucleotide"

There are issues with some of the figure legends: Figures 3-5 have grayscale legends but coloured figure elements, while the shades in Figure 4 are difficult / impossible to differentiate.

Publication Fees:

We recognize that the video files can become quite large, and so to avoid quality loss ASM suggests sending the video file via <https://www.wetransfer.com/>. When you have a final version of the video and the still ready to share, please send it to mSystems staff at mssystemsjournal@msubmit.net.

For mSystems research articles, if you would like to submit an image for consideration as the Featured Image for an issue, please contact mSystems staff at mjournal@msubmit.net.

Sincerely,

Robert Beiko
Editor, mSystems

Journals Department
Fig. S3: Accept
Table S3: Accept
Fig. S4: Accept
Table S4: Accept
Table S6: Accept
Fig. S1: Accept
Table S1: Accept
Table S5: Accept
Table S2: Accept
Fig. S2: Accept